

# New seed coating containing *Trichoderma viride* with anti-pathogenic properties

Sena Turkan[1,2], Agnieszka Mierek-Adamska[1,2], Milena Kulasek[1,2], Wiktoria B. Konieczna[1] and Grażyna B. Dąbrowska[1]

[1] Department of Genetics/Faculty of Biological and Veterinary Sciences, Nicolaus Copernicus University in Toruń, Toruń, Poland
[2] Centre for Modern Interdisciplinary Technologies, Nicolaus Copernicus University in Toruń, Toruń, Poland

Corresponding author
Sena Turkan,
senaturkan@doktorant.umk.pl

## ABSTRACT

**Background:** To ensure food security in the face of climate change and the growing world population, multi-pronged measures should be taken. One promising approach uses plant growth-promoting fungi (PGPF), such as *Trichoderma*, to reduce the usage of agrochemicals and increase plant yield, stress tolerance, and nutritional value. However, large-scale applications of PGPF have been hampered by several constraints, and, consequently, usage on a large scale is still limited. Seed coating, a process that consists of covering seeds with low quantities of exogenous materials, is gaining attention as an efficient and feasible delivery system for PGPF.

**Methods:** We have designed a new seed coating composed of chitin, methylcellulose, and *Trichoderma viride* spores and assessed its effect on canola (*Brassica napus* L.) growth and development. For this purpose, we analyzed the antifungal activity of *T. viride* against common canola pathogenic fungi (*Botrytis cinerea*, *Fusarium culmorum*, and *Colletotrichum* sp.). Moreover, the effect of seed coating on germination ratio and seedling growth was evaluated. To verify the effect of seed coating on plant metabolism, we determined superoxide dismutase (SOD) activity and expression of the stress-related *RSH* (*RelA/SpoT* homologs).

**Results:** Our results showed that the *T. viride* strains used for seed coating significantly restricted the growth of all three pathogens, especially *F. culmorum*, for which the growth was inhibited by over 40%. Additionally, the new seed coating did not negatively affect the ability of the seeds to complete germination, increased seedling growth, and did not induce the plant stress response. To summarize, we have successfully developed a cost-effective and environmentally responsible seed coating, which will also be easy to exploit on an industrial scale.

## INTRODUCTION

In 2021, as many as 828 million people around the world suffered from hunger, and, due to the slow economic recovery from the COVID-19 pandemic, it is predicted that nearly 670 million people will still be facing hunger in 2030 (*Food and Agriculture Organization (FAO), 2022*). Currently, to achieve a high yield, significant inputs of crop protection

chemicals and fertilizers are required. Besides adverse effects on human health, these preparations pollute water, soil, and air and diminish the biological diversity in ecosystems. Therefore, there is an urgent need to develop new agricultural methods to minimize agrochemical use while ensuring production sustainability, food security, environmental responsibility, and cost-effectiveness. The application of plant growth-promoting fungi (PGPF) has been considered a promising approach to replace agrochemicals. Successful application of PGPF in the field is challenging because of the low level of survival, low efficiency of colonization of soil, and short shelf-life of inoculum (*Rojas-Sánchez et al., 2022*). Moreover, it is widely observed that bioformulations beneficial in laboratory conditions do not enhance plant growth and development in field studies (*Mawar, Manjunatha & Kumar, 2021*). The most common methods of application of PGPF are direct foliar, seed, root, and soil inoculation, which might give rise to problems when used on a large scale (*Lopes et al., 2021*). Recently, seed coating has been proposed as an efficient and cost-effective technology for delivering beneficial microorganisms (*Rocha et al., 2019b*; *Ma, 2019*). Seed dressing, film coating, and pelleting are the three major methods of seed coating application and are selected depending on the used chemicals, the form of seed, and selected microbes (*Pedrini et al., 2017*; *Rocha et al., 2019b*). In seed coating, three main types of materials are used: (i) binders, *i.e.*, usually a liquid with adhesive properties, (ii) fillers, *i.e.*, inner material that increases seed size, (iii) active compounds, *e.g.*, microorganisms, micronutrients, and plant growth promoters (*Pedrini et al., 2017*). The coating is performed to enhance seed morphology, to improve handle traits, including seed size and weight, and to deliver active substances (*Pedrini et al., 2017*; *Rocha et al., 2019a*).

Relatively little is known about the possible adverse effects of seed coating on seed metabolism. Reactive oxygen species (ROS) are oxygen-containing radicals commonly considered harmful at high levels that may lead to DNA, protein, and lipid damage. However, when maintained at a moderate level, they play a pivotal role in regulating many processes, including seed germination (*Bailly, 2019*). During seed imbibition, environmental cues are perceived and transduced with the participation of ROS. In permissive conditions, elevated ROS levels trigger germination (*Oracz & Karpiński, 2016*). The task of the cellular antioxidant system is to fine-tune the ROS level to match the current signaling needs. One of the primary antioxidant enzymes is superoxide dismutase (SOD) which catalyzes the conversion of superoxide radical ($O_2 \cdot^-$) into molecular oxygen ($O_2$) and hydrogen peroxide ($H_2O_2$). Since SOD is the first line of defense against enhanced ROS production, the level of SOD activity is often used as a marker of oxidative stress (*Sarker & Oba, 2018*; *Luo et al., 2019*). Recently, much attention has been drawn to the stringent response-related plant *RSH* (*RelA/SpoT* homolog) genes that encode synthases/hydrolases of (pp)pGpp (guanosine tri-, tetra-, and pentaphosphate) called "alarmones", which serve as stress-signaling molecules. The stringent response was observed for the first time over a half-century ago in *Escherichia coli* (*Cashel & Gallant, 1969*), and plant *RSH* genes were first identified in *Arabidopsis thaliana* (L.) Heynh. (*van der Biezen et al., 2000*) and later in other plant species, including *Ipomoea nil* (L.) Roth (syn. *Pharbitis nil* (L.) Choisy) (*Dąbrowska, Prusińska & Goc, 2006*). Plant RSH proteins
have been proven to play a role in stress response to salinity (*Takahashi, Kasai & Ochi, 2004*; *Prusińska et al., 2019*; *Dąbrowska et al., 2021b*), drought (*Takahashi, Kasai & Ochi, 2004*; *Prusińska et al., 2019*), darkness (*Ono et al., 2021*), wounding, UV, heat shock, and pathogens (*Takahashi, Kasai & Ochi, 2004*). *Abdelkefi et al. (2018)* showed that, in *A. thaliana*, the accumulation of alarmones directly correlates with the susceptibility to *turnip mosaic virus* (TuMV) infection. Plant RSHs also play a role in reutilizing micro- and macro-elements from aging organs to seeds (*Boniecka et al., 2017*), photoperiod (*Prusińska et al., 2019*), and the interaction with plant growth-promoting rhizobacteria (*Szymańska et al., 2019*; *Dąbrowska et al., 2021b*).

Canola (*Brassica napus* L.) is, after soybean, the second most important plant oil source used not only for food but also for biofuel production. Canola is sensitive to various environmental stresses, and significant yearly yield loss therefore occurs (*Elferjani & Soolanayakanahally, 2018*). In this study, we aimed to develop a new, biodegradable coating for canola seeds to stimulate germination—one of the most critical stages in plant development that affects all further stages of plant growth and, consequently, yield. Although seed coating as a method of PGPF delivery has been gaining popularity during the last decade, there are still gaps in existing knowledge that prevent this approach being used on an industrial scale. A suitable formulation is challenging to select since the physical and chemical properties of different types of seed coating ingredients may differ depending on other ingredients. Some ingredients promoting the growth of microorganisms might negatively affect plant germination and growth and vice versa. Moreover, seed coating might reduce the shelf life of seeds (*Rocha et al., 2019b*). Since different microbes might react differently with coating materials, more research on different coating formulas is still needed. We created a seed coating consisting of *Trichoderma viride* Pers. spores, chitin as a filler, and methylcellulose as a binder. Fungi belonging to the genus *Trichoderma* have great potential to be used in agriculture because of their ability to stimulate plant growth and development, including in adverse environmental conditions (*Lorito & Woo, 2015*; *Contreras-Cornejo et al., 2016*; *Al-Ani, 2018*; *Macías-Rodríguez et al., 2020*) *via* various mechanisms including modifying the rhizosphere, modulating root architecture, increasing the availability of nutrients, and producing specific growth and development-promoting compounds (*Guzmán-Guzmán et al., 2019*; *Mastan et al., 2021*; *Antoszewski, Mierek-Adamska & Dąbrowska, 2022*). In addition, *T. viride* can grow on polymeric materials and possibly can degrade them (*Dąbrowska et al., 2021a*). We chose spores of *T. viride* as the active compound of the seed coating because our recent study showed the positive effect of this species on the development of canola (*Znajewska, Dąbrowska & Narbutt, 2018*). In addition, fungi belonging to *Trichoderma* might control populations of plant pathogens (*Rodríguez et al., 2021*) *via* the secretion of hydrolytic enzymes (*Pérez et al., 2002*). Therefore, we hypothesized that the tested *T. viride* strains possess anti-fungal activity and verified *T. viride* activity against plant pathogen fungi, *i.e.*, *Botrytis cinerea* Pers., *Fusarium culmorum* (Wm.G. Sm.) Sacc., and *Colletotrichum* sp. Those fungi are the most significant plant pathogens worldwide, causing diseases in a wide range of hosts, including cereals, legumes, vegetables, and fruit trees (*Kthiri et al., 2020*) and additionally posing serious health risks (*Juergensen & Madsen, 2009*; *Shivaprakash*

*et al., 2011*). They are present in food and feed made from contaminated cereal crops (*Błaszczyk et al., 2017*). Large-scale disease control techniques are implemented using chemical fungicides, which are expensive, harmful to living organisms, and environmentally irresponsible. Spraying with a spore suspension of *Trichoderma* fungi has a well-documented protective anti-fungal effect, *e.g.*, on soybean (*John et al., 2010*), chickpea (*Pradhan et al., 2022*) or cocoa (*Seng et al., 2014*). Recently, *Trichoderma*-enriched seed coatings have been developed for many plant seeds (*Cortés-Rojas et al., 2021*), and the coating allows beneficial microorganisms to colonize the roots at an early stage of growth (*Tavares et al., 2013*; *Ben-Jabeur et al., 2019*). Secondly, we checked the effect of seed coating on canola seed germination and seedling growth. We aimed to show that chitin could be used as a filler instead of commonly used chitosan. Although chitin has lower solubility than chitosan, chitosan is less stable, more hydrophilic, and more sensitive to changes in pH. Moreover, chitosan can form complexes with metals, including microelements, thereby reducing their bioavailability for plants and fungi. Chitosan also has stronger antimicrobial activity than chitin, which is a disadvantage for microbe-containing seed coatings (*Rinaudo, 2006*; *Aranaz et al., 2009*). From the industrial point of view, the coating materials should be low-cost, and chitin is significantly less expensive than chitosan. Lastly, we hypothesized that the newly developed seed coating would be inert to seed metabolism, and thus we assessed the expression of the *RSH* genes and the activity of SOD as markers of stress in plants.

## MATERIALS AND METHODS

### Microorganisms

The saprophytic fungi used in this study were obtained from the culture collection of the Department of Microbiology (GenBank NCBI accession numbers: OL221594.1–*T. viride* strain I (TvI) and OL221590.1–*T. viride* strain II (TvII), and from the culture collection of the Department of Environmental Microbiology and Biotechnology (The Bank of Pathogens, Institute of Plant Protection in Poznań; Faculty of Forestry and Wood Technology Poznań University of Life Sciences; *B. cinerea* - 873, *Colletotrichum* sp.- 1202, and *F. culmorum*- 2333), Faculty of Biological and Veterinary Sciences, Nicolaus Copernicus University in Toruń. The culture of fungi stored in agar slopes at 4 °C was transferred to a solid PDA (Potato Dextrose Agar, Difco, US) medium and incubated at 23 °C for 7 days. Fungal spores were collected with a sterile cotton-tipped applicator and suspended in sterile water.

### *In vitro* antagonistic activity of *T. viride* strains against plant pathogens

*In vitro* antagonist activity of *T. viride* against plant pathogens (*B. cinerea*, *Colletotrichum* sp., and *F. culmorum*) was evaluated using the dual culture technique according to *Li et al. (2016)* and *Kunova et al. (2016)*. Mycelial discs of 5 mm diameter of 1-week-old *T. viride* and 1-week-old plant pathogens were placed on the opposite sides of Petri dishes (maintaining a distance of 6 cm between discs) containing Czapek-Dox agar (2.0 g NaNO$_3$, 1.0 g K$_2$HPO$_4$, 0.5 g MgSO$_4$ × 7 H$_2$O, 0.5 g KCl, 0.01 g FeSO$_4$ × 7 H$_2$O, 20 g glucose, 14 g agar, 1 g yeast extract dissolved in 1,000 mL distilled water). Control plates contained only

**Table 1 Experimental variants tested in this study.**

| Variant | Binding agent | Filler | Active compound | Seed weight (mg) |
|---|---|---|---|---|
| Uncoated seed | – | – | – | 5.142 ± 0.291b |
| M | + | – | – | 5.863 ± 0.169a |
| C | – | + | – | 5.301 ± 0.300b |
| M+C | + | + | – | 5.902 ± 0.117a |
| TvI | – | – | + | 5.220 ± 0.290b |
| TvII | – | – | + | 5.219 ± 0.285b |
| M+C+TvI | + | + | + | 5.903 ± 0.358a |
| M+C+TvII | + | + | + | 5.905 ± 0.197a |

Note:
The full seed coating (M+C+TvI, M+C+TvII) consists of *Trichoderma viride* spores (active compound), methylcellulose (binding agent), and chitin (filler). Uncoated seeds, seeds coated with methylcellulose (M), chitin (C), methylcellulose and chitin (M+C), and *T. viride* spores (TvI/TvII) served as control. The mass of one seed for each experimental variant was determined based on the weight of 100 random seeds. Values are the mean ± SD ($n = 4$). Different letters indicate significant differences between variants (ANOVA with Tukey's *post-hoc* test, $p < 0.05$).

mycelial discs of *B. cinerea*, *Colletotrichum* sp., and *F. culmorum*. The plates were incubated at 25 °C for 6 days in the dark, and the diameter of mycelia was measured. The pathogen growth inhibition was calculated according to the following formula:

$$\text{Inhibition}(\%) = [(C - T)/C] \times 100 \tag{1}$$

where C is the radial growth of plant pathogen (mm) when grown without *T. viride* (control), and T is the radial growth of plant pathogen (mm) in the presence of *T. viride* strains.

## Seed coating

Seeds of *B. napus* 'Karo' were obtained from Plant Breeding Strzelce Ltd., Co. (Strzelce, Poland; IHAR-PIB Group). 'Karo' is a spring open-pollinated variety of canola, registered in Poland in 2016. This cultivar produces a high yield, seeds contain a high level of oil, and the plant is resistant to fungal pathogens. The seeds were surface sterilized with a mixture of 30% hydrogen peroxide and 96% ethanol (1:1, v:v) for 3 min and then rinsed at least six times with sterile distilled water. The complete coating mixture (C+M+TvI, C+M+TvII) consisted of a solution of 0.5% chitin dissolved in methylcellulose (final concentration of 2.5%) and *T. viride* spore suspension at a final concentration of $10^5$ spores mL$^{-1}$.
For control, the following variants were used: (i) 0.5% chitin (C) that was dissolved in 1% acetic acid (the pH of the solution was adjusted to 6.0 using 1% NaOH), (ii) 2.5% methylcellulose (M), (iii) the mixture of methylcellulose (final concentration 2.5%) and chitin (final concentration 0.5%) (M+C), (iv) *T. viride* spores at a final concentration of $10^5$ spores mL$^{-1}$ in distilled sterile water (TvI/TvII). In each variant, 100 mg of canola seeds were incubated with 5 mL of mixture with shaking at 180 rpm for 15 min at room temperature. The summary of experimental variants and the means mass of seed tested in this study are presented in Table 1.

## Impact of seed coating on *B. napus* germination and seedling growth

For the germination test, 25 seeds were placed in glass Petri dish (90 mm) with filter paper moistened with distilled water (conductivity < 0.08 µS cm$^{-1}$). Subsequently, the dishes were placed in a growth chamber, temperature regulated to 25 °C (at 16 h light and 8 h dark cycles), with supplemental lighting to maintain a light intensity, PAR = 100 µmol photons m$^{-2}$ s$^{-1}$. The seed germination was monitored for 7 days. To assess the rate of germination, three different germination parameters were calculated. FGP and MGT were calculated following the formula by *Ranal & Santana (2006)*, while IGV was calculated using the formula reported by *Khan & Ungar (1984)*:

Final germination percentage, $FGP = 100(N/S)$       (2)

where N is the number of total seeds completed germination at the end of the experiment and S is the number of initial seeds used;
Index of germination velocity (modified Timson's index),

$IGV = \Sigma G/t$       (3)

where G is the percentage of seed germination at 1-day intervals, and t is the total germination period;
Mean germination time,

$MGT = (N1T1 + N2T2 + \ldots + NxTx)/(N1 + N2 + \ldots + Nx)$       (4)

where N is the germination count on any counting period, and T is the time point in days until the last day (x).

To assess the growth, we measured the lengths of roots and shoots and the fresh and dry biomass of 100 seven-day-old seedlings. The plants were carefully separated into shoots and roots using a razor blade, and the lengths of roots and shoots were measured using a calibrated ruler. To assess the dry mass, samples were dried and weighed using an MA 50 moisture analyzer (Radwag, Radom, Poland).

## Determination of SOD activity

For SOD activity analysis plants were grown as described above. Protein extracts were prepared according to *Rusaczonek et al. (2015)*. Briefly, 6-day-old seedlings were ground in liquid nitrogen, and then ~50 mg of the grounded tissue was suspended in 1 mL of ice-cold protein extraction buffer (100 mM tricine, 3 mM MgSO$_4$, 3 mM EGTA, 1 mM DTT, pH 7.5). Following the incubation on ice for 15 min, samples were centrifuged (4 °C, 20 min, 13,000 rcf). The obtained supernatant was used to determine total soluble protein concentration using PierceTM Coomassie (Bradford) Protein Assay Kit (Merck, Darmstadt, Germany) according to the manufacturer's protocol, with bovine serum albumin as a protein standard. SOD activity in the protein extract was measured according to *Beauchamp & Fridovich (1971)*, adapted by *Rusaczonek et al. (2015)*. Briefly, 100 µL of the working solution (0.1 M phosphate buffer pH 7.5, 2.4 µM riboflavin, 840 µM NBT, 150 mM methionine, 12 mM Na$_2$EDTA mixed in the ratio: 8:1:1:1:1 (v/v/v/v/v)) was added to 2 µL of the protein extract in a 96-well transparent microplate in three technical

**Table 2 Primers used for *Brassica napus* L. *RSH* genes expression analysis.**

| Target gene | Gene ID | Name of oligomer | Nucleotide sequence (5′ → 3′) |
|---|---|---|---|
| *BnRSH1* | 106399012 | Sense<br>antisense | 5′- GGAGGTTCAGATCAGAACGG - 3′<br>5′- CCATTCACCTTCGCTGCTAC - 3′ |
| *BnRSH2* | 111206471 | Sense<br>antisense | 5′- GCAAGATGTTGAAGAATCTAACG - 3′<br>5′- GCACAGACATCTTGTCATTTTCG - 3′ |
| *BnRSH3* | 106431664 | Sense<br>antisense | 5′- CCGAAACTTTCCGATTTCAA - 3′<br>5′- TCGTAGTCAACGCACGAGTC - 3′ |
| *BnCRSH* | 106439579 | Sense<br>antisense | 5′- ACGTTCTCGGTCTCCGTGTC - 3′<br>5′- CGCTTTCGGCTTAGCGATGT - 3′ |
| *BnUBC9* | 106376144 | Sense<br>antisense | 5′- GCATCTGCCTCGACATCTTGA - 3′<br>5′- GACAGCAGCACCTTGGAAATG - 3′ |

replicates for each of three biological replicates. The reaction was set in two identical plates, one illuminated with 400 µE warm white LED light for 16 min, and the other (blank) was kept in darkness. Then absorbance at 560 nm was measured. One unit of SOD activity was defined as the amount of enzyme required for 50% inhibition of NBT reduction in 1 min in the assay conditions.

## Expression analysis of *BnRSH* genes

Real-time PCR (qPCR) assays were performed to evaluate the effects of seed coating on *BnRSH1, BnRSH2, BnRSH3*, and *BnCRSH* expression. Total RNA was isolated from 6-day-old seedlings using the RNeasy Plant Mini Kit (Qiagen, Hilden, Germany) according to the manufacturer's protocol. The integrity and quantity of RNA were analyzed by spectrophotometric measurement and electrophoresis in 1% agarose gel in 1x TAE buffer stained with ethidium bromide. To remove genomic DNA contamination, RNA was treated with 1 U of DNase I (Thermo Fisher Scientific, Waltham, MA, US). The DNase I was heat-inactivated in the presence of 5 mM EDTA. The cDNA was synthesized from 1.5 µg of total RNA using both 250 ng oligo (dT)$_{20}$ primer and 200 ng random hexamers with NG dART RT Kit (EURx, Gdańsk, Poland). The reaction was performed at 25 °C for 10 min, followed by 50 min at 50 °C.

The qPCR reaction mixture included 4 µL of 2-fold diluted cDNA, gene-specific primers at a final concentration of 0.5 µM each, and 5 µL of LightCycler 480 SYBR Green I Master (Roche, Penzberg, Germany) in a total volume of 10 µL. Primers for *BnRSH* are listed in Table 2. The reaction was performed in three technical replicates for three biological replicates in LightCycler 480 Instrument II (Roche, Penzberg, Germany). PCR conditions were as follows: 95 °C for 5 min for initial denaturation, and 40 cycles of 95 °C for 10 s, 59 °C for 20 s, and 72 °C for 20 s. The fluorescence signal was recorded at the end of each cycle. To verify the specificity of the PCR reaction melt curve analysis was used (55 °C to 95 °C at a ramp rate of 0.11 °C/s). The relative gene expression was calculated using LightCycler 480 Software version 1.5.1.62 (Roche, Penzberg, Germany).

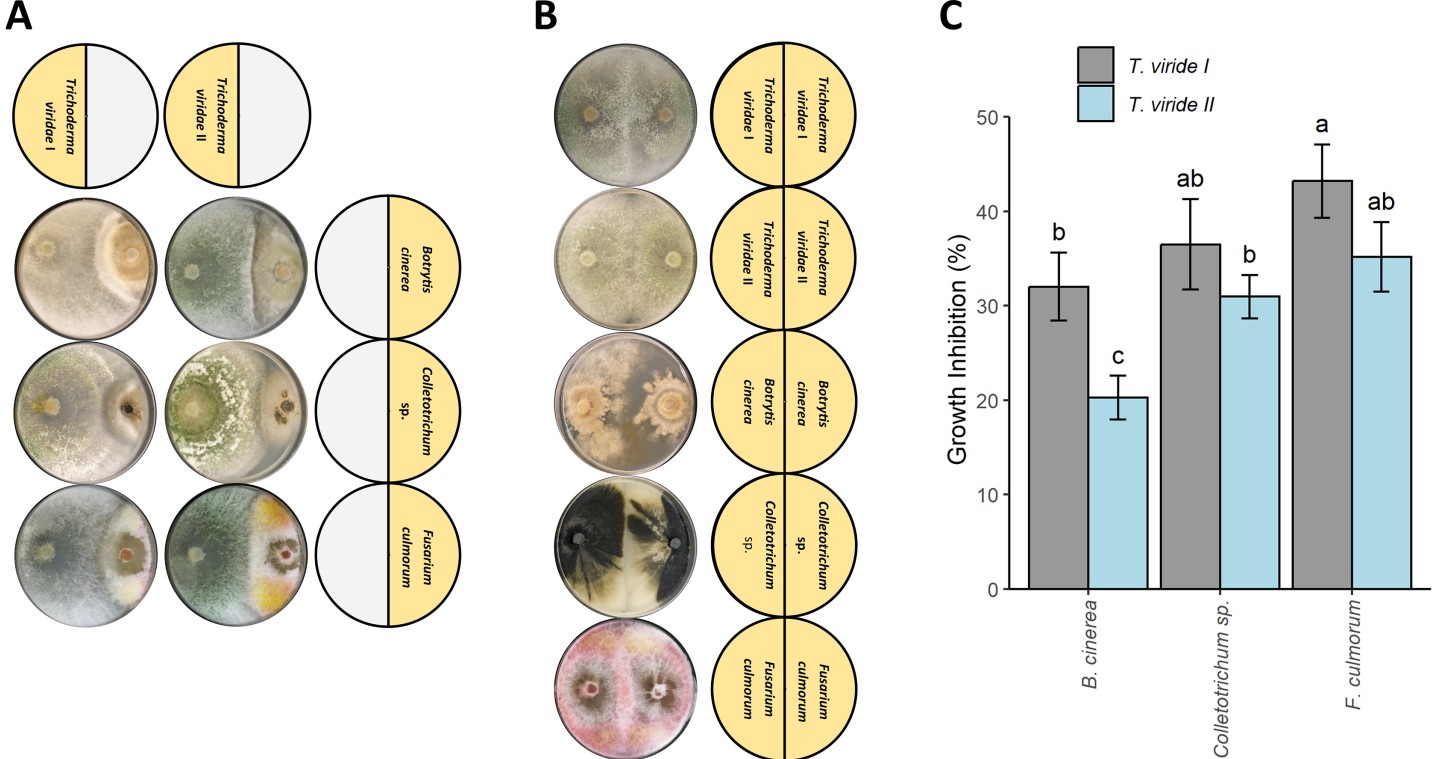

**Figure 1 Effect of *Trichoderma viride* isolates (Tv I–II) on the growth of plant pathogens *Botritis cinerea*, *Colletotrichum* sp., and *Fusarium culmorum*.** (A) Co-culture experiment showing the antagonism of *T. viride* I and II (left side of Petri dish) against plant pathogens: *B. cinerea*, *Colletotrichum* sp., and *F. culmorum* (right side of Petri dish). (B) Control showing the growth of tested *T. viride* strains and pathogens. (C) Percentage of inhibition of *T. viride* isolates (Tv I–II) against the plant pathogens *B. cinerea*, *Colletotrichum* sp., and *F. culmorum*.

## Statistical analyses

The homogeneity of variances and normality of the data was tested with the Bartlett and Shapiro–Wilk tests, respectively. To test the significance of the differences in means, the ANOVA test was applied, followed by Tukey's *post-hoc* test when data met the normality assumption and the Mann–Whitney *post-hoc* test when data did not meet the normality assumption. Data were visualized using the 'ggplot2' R package.

## RESULTS

### The effect of *T. viride* on the growth of pathogenic fungi

Both tested *T. viride* isolates (*T. viride* I and *T. viride* II) significantly inhibited the growth of all plant pathogens analyzed in this study *i.e.*, *B. cinerea*, *Colletotrichum* sp., and *F. culmorum* (Figs. 1A and 1C). *T. viride* I showed a percentage of inhibition in the range of 33.03–43.21% whereas *T. viride* II showed a percentage of inhibition in the range of 20.27–35.19% (Fig. 1C). *T. viride* I suppressed the growth of all analyzed pathogens more than *T. viride* II. The strongest inhibition was observed for *T. viride* I and *F. culmorum*, *i.e.*, 43.21% of inhibition, while the weakest inhibitory effect was observed for *T. viride* II and *B. cinerea*, *i.e.*, 20.27%. In fact, both tested *T. viride* strains the least effectively inhibited the growth of *B. cinerea* (Fig. 1C). Interestingly, the discoloration of *F. culmorum* mycelia

**Table 3** Final germination percentage (FGP), index of germination velocity (IGV), and mean germination time (MGT) of *Brassica napus* L. seeds depending on type of seed coating.

| Treatments | FGP (%) | IGV (a.u.) | MGT (day) |
|---|---|---|---|
| Control | 85 ± 13.78a | 51.6 ± 7.69a | 4.76 ± 0.06a |
| M+C+TvI | 98 ± 4.08a | 62.8 ± 6.7 a | 4.65 ± 0.12a |
| M+C+TvII | 98 ± 4.08a | 64.2 ± 5.86a | 4.64 ± 0.12a |
| Methylcellulose | 91 ± 9.83a | 53.0 ± 10.86a | 4.75 ± 0.12a |
| Chitin | 86 ± 8.16a | 52.1 ± 8.93a | 4.73 ± 0.08a |
| Methylcellulose + Chitin | 90 ± 10.95a | 52.8 ± 9.73a | 4.71 ± 0.09a |
| TvI | 95 ± 5.48a | 60.9 ± 2.81a | 4.63 ± 0.05a |
| TvII | 93 ± 8.16a | 62.1 ± 4.12a | 4.63 ± 0.04a |

Note:
Values are mean ± SD (*n* = 4). Different letters indicate significant differences between groups (ANOVA with Tukey's *post-hoc* test, *p* < 0.05). Uncoated—control seeds, M+C+TvI—methylcellulose-chitin-*Trichoderma viride* I treated seeds, M+C+TvII—methylcellulose-chitin-*T. viride* II treated seeds, M—methylcellulose treated seeds, C—chitin treated seeds, M+C—methylcellulose–chitin treated seeds, TvI —*T. viride* I treated seeds, and TvII —*T. viride* II treated seeds.

upon contact with *T. viride* was observed, *i.e.*, pink *F. culmorum* mycelia in control (Fig. 1B) changed into white/yellow when grown together with *T. viride* (Fig. 1A). The two-way ANOVA revealed no statistically significant interaction between the effects of *T. viride* strains and pathogens (F-value = 1.147, *p*-value = 0.350043). However, simple main effects analyses showed that both *T. viride* strains and pathogens did have a statistically significant effect on inhibition (for *Trichoderma*: F-value = 24.967, *p*-value = 0.000311, for pathogens: F-value = 20.082, *p*-value = 0.000148).

## The effect of seed coating on seed germination and seedling growth

To verify the beneficial influence of the developed *T. viride*-containing seed coating on plant development and growth, we performed the germination test (Table 3) and measured the growth dynamics of seedlings (Fig. 2). Final germination percentage (FGP), which reflects only the final ratio of seeds that completed germination, was the highest for seeds with the complete coating (*i.e.*, methylcellulose + chitin + *T. viride* spores), *i.e.*, 98%. In contrast, the lowest value of FGP was observed for control seeds, *i.e.*, 85% (Table 3). However, those differences were not statistically significant. FGP does not provide information about the speed or uniformity of germination; therefore, the index of germination velocity (IGV) and the mean germination time (MGT) were calculated. IGV, which indicates the rapidity of germination, was the highest for seeds with complete seed coating (*i.e.*, methylcellulose + chitin + *T. viride* spores) and the lowest for control seeds. Interestingly, the IGV value for seeds with complete coating was higher in comparison to seeds inoculated only with *T. viride* spores, which showed that other ingredients of seed coating, *i.e.*, methylcellulose and chitin, did not negatively affect the germination-promoting ability of *T. viride* (Table 3). MGT reflects the time needed for a group of seeds to complete germination and focuses on the day when most of the germination was completed. Faster germination was observed in seeds with complete coating and seeds inoculated with *T. viride* spores, whereas, for other experimental variants, the MGT was the same as for control seeds (Table 3). To assess the effect of seed

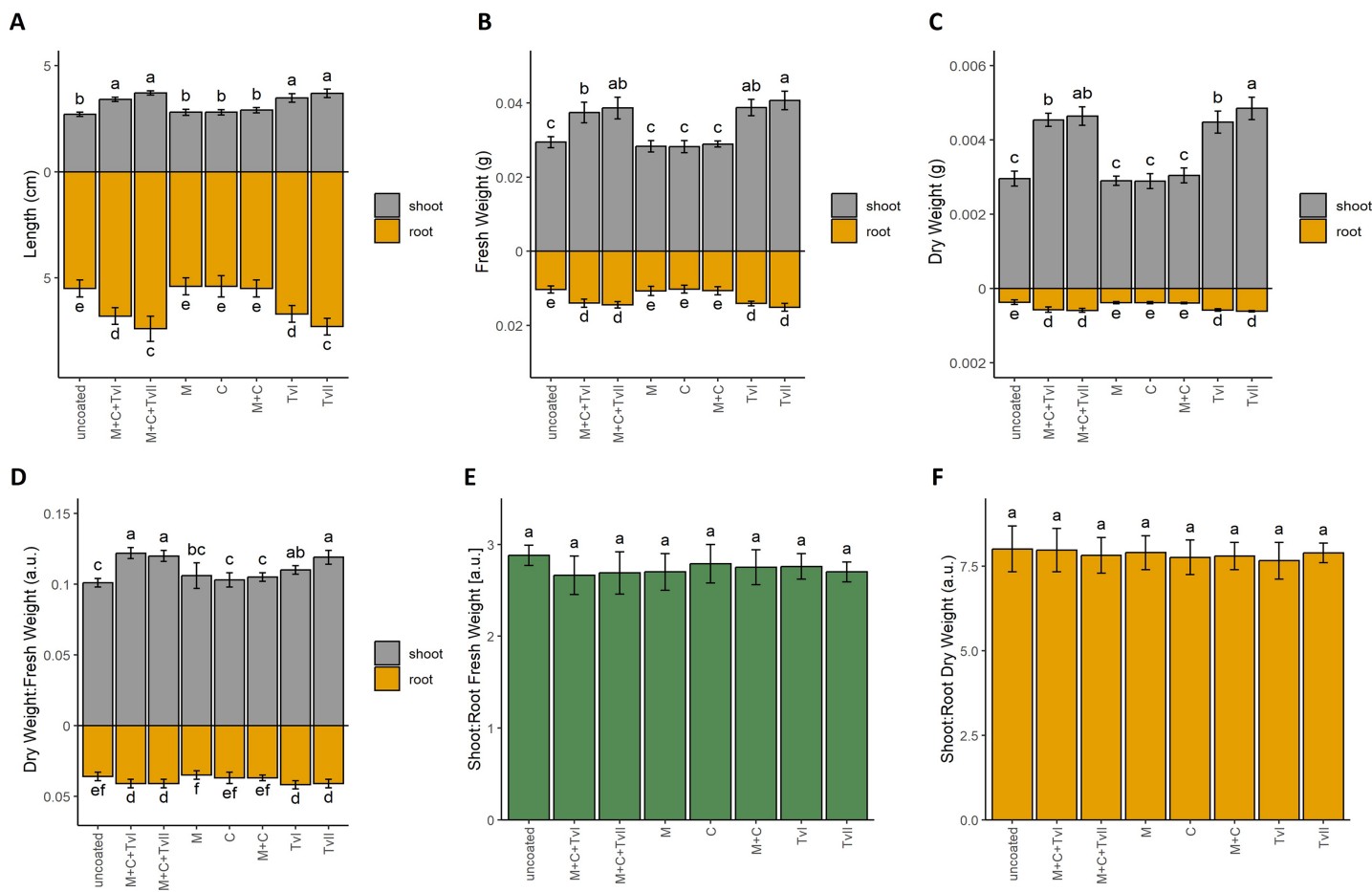

**Figure 2** Length (A), fresh weight (B), dry weight (C), and dry weight:fresh weight ratio (D) of shoot and roots, and fresh weight (E) and dry weight (F) shoot:root ratio of *Brassica napus* L. seedlings depending on type of seed coating. Values are mean ± SD (*n* = 4). Different letters indicate significant differences between groups (ANOVA with Tukey's *post-hoc* test and Mann–Whitney *post-hoc* test, *p* < 0.05). Uncoated—control seeds, M+C+TvI—methylcellulose-chitin-*Trichoderma viride* I treated seeds, M+C+TvII—methylcellulose-chitin-*T. viride* II treated seeds, M—methylcellulose treated seeds, C—chitin treated seeds, M+C—methylcellulose–chitin treated seeds, TvI—*T. viride* I treated seeds, and TvII—*T. viride* II treated seeds.

coating on further seedling growth, we measured the length of roots and hypocotyls of 6-day-old seedlings (Fig. 2A). The seedlings that grew from seeds coated with a complete coating (*i.e.*, methylcellulose + chitin + *T. viride* spores) displayed longer shoots and roots than those from uncoated seeds. Seedlings that grew from seeds inoculated with spores of *T. viride* had also significantly longer shoots and roots compared to seedlings that grew from uncoated seeds (Fig. 2A), which showed that chitin and methylcellulose did not affect the ability of *T. viride* to promote the growth of canola seedlings.

Fresh (Fig. 2B) and dry (Fig. 2C) weight, as well as dry matter content (Fig. 2D), of 6-day-old canola seedlings were higher for seedlings that grew from seeds with the complete coating (*i.e.*, methylcellulose + chitin + *T. viride* spores) and from seeds inoculated with *T. viride* spores when compared to uncoated seeds and other control variants (*i.e.*, chitin, methylcellulose, and chitin + methylcellulose). The fungus' plant growth promoting ability was thus unaffected by the filler and binder in seed coating.

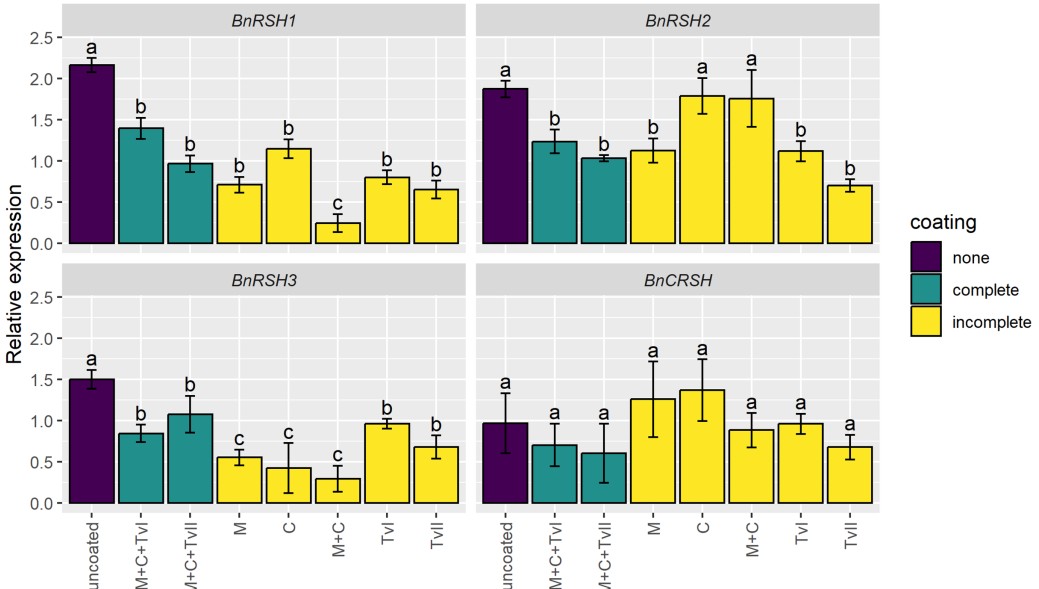

**Figure 3 Relative expression level of *BnRSH* genes in 6-day-old *Brassica napus* L. seedlings germinated from uncoated (control) or coated seeds.** Values are mean ± SD (*n* = 4). Different letters indicate significant differences between groups (ANOVA with Tukey's *post-hoc* test, *p* < 0.05). Uncoated—control seeds, M+C+TvI—methylcellulose-chitin-*Trichoderma viride* I treated seeds, M+C +TvII—methylcellulose-chitin-*T. viride* II treated seeds, M—methylcellulose treated seeds, C—chitin treated seeds, M+C—methylcellulose–chitin treated seeds, TvI—*T. viride* I treated seeds, and TvII— *T. viride* II treated seeds.

In each case, the highest biomass was observed for seedlings inoculated with spores of *T. viride* II (*i.e.*, 1.5-fold higher for fresh root weight, 1.7-fold higher for fresh shoot weight, 1.4-fold higher for dry root weight, and 1.6-fold higher for dry shoot weight than seedlings that grew from uncoated seeds). The biomasses of seedlings that grew from seeds with a complete coating containing spores of *T. viride* II, chitin, and methylcellulose were only slightly lower than the previously mentioned (*i.e.*, 1.4-fold higher for fresh root weight, 1.6-fold higher for fresh shoot weight, 1.3-fold higher for dry root weight, and 1.6-fold higher for dry shoot weight than seedlings that grew from uncoated seeds). The comparison of shoot:root ratio (Figs. 2E and 2F) showed that seedlings that grew from tested seed coatings had the same proportion of shoots and roots. These results suggest that seed coating similarly promoted the growth of both roots and shoots in all tested variants.

## The effect of seed coating on *BnRSH* gene expression

To assess whether the seed coating induces stress-related genes, the expression of genes encoding synthases and/or hydrolases of alarmones, *i.e.*, *BnRSH*s, was evaluated in 6-day-old seedlings grown from coated and uncoated seeds. The expression of no *BnRSH* genes was induced by seed coating (Fig. 3). The expression of *BnCRSH* was not affected by seed coating, whereas the expression of *BnRSH1–3* was down-regulated in seedlings grown from coated seeds compared to seedlings that grew from uncoated seeds. The transcript levels of *BnRSH1* and *BnRSH3* were significantly reduced in seedlings grown from all coated seeds, but the lowest level of expression (*i.e.*, almost nine-times lower than in

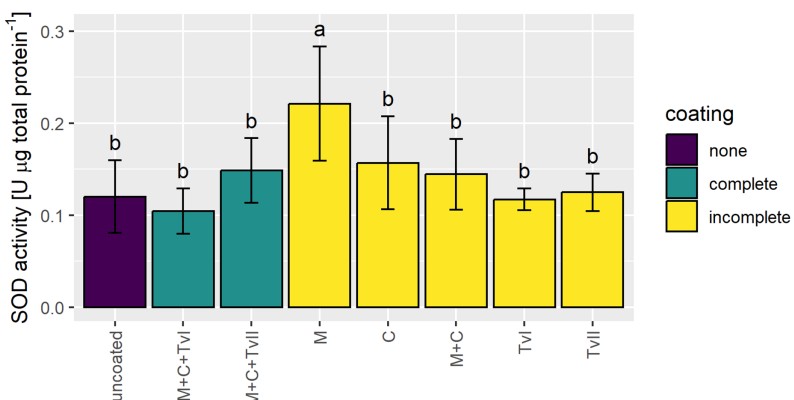

**Figure 4 Activity of SOD in 6-day-old *Brassica napus* L. seedlings germinated from uncoated (control) and coated seeds.** Values are mean ± SD ($n = 4$). Different letters indicate significant differences between groups (ANOVA with Tukey's *post-hoc* test, $p < 0.05$). Uncoated—control seeds, M+C+TvI—methylcellulose-chitin-*Trichoderma viride* I treated seeds, M+C+TvII—methylcellulose-chitin-*T. viride* II treated seeds, M—methylcellulose treated seeds, C—chitin treated seeds, M+C—methylcellulose–chitin treated seeds, TvI—*T. viride* I treated seeds, and TvII —*T. viride* II treated seeds.

control seedlings) was observed for *BnRSH1* in seedlings grown from seeds coated with the mixture of chitin and methylcellulose (Fig. 3).

## The effect of seed coating on SOD activity

Superoxide dismutase is a ubiquitous metalloenzyme that comprises the first level of protection against reactive oxygen species (ROS) and is widely used as a biochemical marker for abiotic stress tolerance in plants (*Szechyńska-Hebda et al., 2007*; *Berwal & Ram, 2019*). Therefore, to verify whether the seed coating developed in this study induced stress in plants, the activity of SOD was measured in 6-day-old seedlings grown from uncoated and coated seeds. The SOD activity significantly increased only in seedlings grown from seeds coated with methylcellulose and was not affected in seedlings grown from other variants of coated seeds in comparison to seedlings grown from uncoated seeds (Fig. 4).

## DISCUSSION

Over the last decades, several biotechnological and agricultural advances have been made in developing new stress-resistant varieties, varieties producing higher yields, and bioinoculants improving plant growth (*Gao, 2021*). However, there is still room for improvement in the agricultural sector. Since many countries have banned genetically modified crops from the market, one of the most promising techniques for yield improvement is artificial seed covers created to aid plant development (*Rihan et al., 2017*). Nowadays, they are made mostly from synthetic materials that are expensive and non-degradable (*Pirzada et al., 2020*; *Britt, 2021*; *Zia et al., 2021*; *Sohail et al., 2022*). As seeds complete germination and young seedlings break through the soil and expand upwards, the plastic capsule suppresses the vegetation underneath (*Bosker et al., 2019*).

## The new seed coating has the potential to improve plant immunity to fungal pathogens

In this study, we have demonstrated the ability of analyzed strains of *T. viride* to suppress the growth of the three plant pathogens (Fig. 1). Similar levels of antagonistic potential between *T. viride* and *F. culmorum* were also observed in other studies (*Modrzewska et al., 2022*). We report here the first quantitative analyses concerning *B. cinerea* and *Colletotrichum* sp. For *B. cinerea*, similar (~30%) inhibition was observed when co-cultured with spore suspension *Trichoderma harzianum* Rifai (*Geng et al., 2022*). For *T. viride* and *Colletotrichum* sp. inhibition, the data published so far is descriptive and image-based only (*Bankole & Adebanjo, 1996*).

A great body of evidence suggests *Trichoderma* treatment contributes to the defense of plants against pathogens far beyond germination. Biopriming of tomato seeds with *T. asperellum* increased the accumulation of total phenol and antioxidant enzyme activities in plants, consequently, and induced resistance against Fusarium Wilt disease caused by *Fusarium oxysporum* f. sp. *lycopersici* (Sacc.) W.C. Snyder & H.N. Hansen (*Singh et al., 2020*). Isolates of *T. harzianum* also promoted growth and systemic protection against downy mildew in a highly susceptible sunflower cultivar (Morden) (*Nagaraju et al., 2012*). Biopriming of durum wheat (*Triticum durum* L.) seeds with *Trichoderma* strains enhanced the systemic resistance against *Fusarium* crown rot caused by *Fusarium culmorum* while also promoting growth (*Kthiri et al., 2020*). Treatment of cowpea seeds with *T. viride* increased the immunity against *Colletotrichum truncatum* (Schwein.) Andrus & W.D. Moore; however, the treatment had to be repeated on a 2-week basis (*Bankole & Adebanjo, 1996*). On this basis, we assume that the *in-vitro* inhibition observed in our study will translate into a similar effect *in planta*, significantly contributing to the improvement of its resistance to the studied pathogens.

It was recognized that the inhibitory effect of *T. viride* and the pathogens probably involves the activation of various mechanisms. First of all, *Trichoderma* spp. have been regarded as necrotrophic mycoparasites, and this lifestyle is supported by the enzymes that break down chitin (*Ihrmark et al., 2010*), reviewed in (*Mukherjee et al., 2022*). Other mechanisms may involve the degradation of pathogen cell walls, production of antibiotics, and competition for nutrients (*Sivan & Chet, 1989*; *Sarrocco et al., 2009*) and ecological niches (*Chet & Inbar, 1994*; *Vinale et al., 2008*; *Vos et al., 2015*; *Waghunde, Shelake & Sabalpara, 2016*; *Oszust, Cybulska & Frąc, 2020*). Recently, *T. harzianum* S. INAT was found to induce the systemic resistance of durum wheat against foot crown rot disease caused by *Fusarium* (*Kthiri et al., 2020*). Significantly, several *Trichoderma* isolates were found to be capable of detoxifying zearalenone, a mycotoxin produced by some *Fusarium* species that contaminate grains (*Tian et al., 2018*), which suggests an additional potential benefit of the new seed coating. However, this needs to be verified in the future.

## The new materials for seed coating promote plant growth

Our results showed that both tested strains of *T. viride* inoculated separately and as a component of seed coating did not impair the ability of the seeds to complete germination (Table 3), and both strains significantly enhanced seedling development (Fig. 2).

The impact of seed treatment with spores of various *Trichoderma* species on germination and growth seems to be plant-species-dependent. Some studies indicate no effect of these fungi on germination (*Lustosa et al., 2020*). However, most of the data show the positive effect of germination to mature plants. In tomato (*Solanum lycopersicum* L.), biopriming with *Trichoderma asperellum* Samuels, Lieckf. & Nirenberg increased seed germination and plant height (*Singh et al., 2020*). In maize (*Zea mays* L.), single inoculation improved the ability of seeds to complete germination and vigor, field emergence, as well as plant height and seed mass (*Nayaka et al., 2010*). In wheat, seed biopriming with *T. harzianum* induced significant effects, specifically increased leaf area, ear length, ear weight, test weight, and grain yield, while reducing chemical fertilization (*Meena et al., 2017*). Similarly, pea seed biopriming with the *T. asperellum* strain BHUT8 effectively increased the length of the shoot and root, number of leaves, and fresh weight of the shoot and root compared to the control (*Singh et al., 2016*). Higher seed ability to complete germination, seedling establishment, and shoot length were observed by *Piri et al. (2019)* following the bio-priming of cumin seeds with *T. harzianum*. In soybean (*Glycine max* L.), seed biopriming with *T. viride* strain BHU-2953 increased root length and phosphorus (P) uptake and thus reduced demand for P-fertilizer (*Paul & Rakshit, 2021*). In sunflower, *T. harzianum* biopriming led not only to higher germination rate, vigor index, and height but also accelerated maturation (*Nagaraju et al., 2012*). In rice, seed biopriming with *Trichoderma* strains improved seed vigor, germination, chlorophyll content, and plant growth and additionally enhanced straw degradation capacity (*Swain et al., 2021*). Therefore, it is apparent that *Trichoderma* biopriming has the potential to benefit crop growth and development beyond germination, and this phenomenon would be seen in mature bio-primed rapeseed plants.

Canola seedlings showed significantly longer shoots and roots in all variants containing *T. viride* (Fig. 2A). Interestingly, *Trichoderma atroviride* P. Karst. was proven to decelerate *A. thaliana* root elongation by rhizosphere acidification (*Pelagio-Flores et al., 2017*). The observed effect seems to be specific to *Trichoderma*–plant species duets, and in *T. viride*–canola, this pathway is probably not induced. This is supported by the study of *Nieto-Jacobo et al. (2017)*, who showed the opposite effects of two *Trichoderma* species on *A. thaliana*. Moreover, the effect of *T. asperellum* LU1370 depended on growth conditions: in the soil, *A. thaliana* exhibited dwarfism, whereas on agar, its growth accelerated (*Nieto-Jacobo et al., 2017*). Also, fresh and dry weights of both roots and shoots were significantly higher in all variants containing *T. viride* spores (Figs. 2B and 2C). *Trichoderma* may induce local or systemic plant resistance through salicylic acid (SA), jasmonic acid (JA), and/or auxin pathways during interaction with plants (*Nawrocka & Małolepsza, 2013*). The mechanism of the observed growth promotion is probably *via* the auxin pathway, a key phytohormone that orchestrates plant growth and development. *Gravel, Antoun & Tweddell (2007)* found that IAA synthesis by a *Trichoderma* sp. was correlated with tomato growth. Since IAA may differently influence various tissues, one of the possible explanations for the phenomenon observed in our study is that, in shoots, both cell division and elongation are induced, and, in roots, only cell division is accelerated. This hypothesis needs to be verified by histological and immunohistochemical analyses.

*Trichoderma* spp. might also boost rapeseed growth through other mechanisms, including the production of volatile organic compounds (*Neik et al., 2020*) and various secondary metabolites (*Vinale et al., 2009*). Moreover, *Trichoderma* strains may be capable of colonizing the roots during early growth of seedlings and hence facilitate early development through enhancing nutrient absorbance (*Saba, 2012*; *Lutts et al., 2016*; *Ben-Jabeur et al., 2019*; *Kthiri et al., 2020*). In addition, these fungi might reduce the activity of harmful root microflora and deactivate toxins in the root area, resulting in improved root growth (*Roberti et al., 2008*).

## The developed seed coating does not induce stress in plants

Based on available results, it was hypothesized that plant RSH proteins might play a role in many physiological processes, including germination and plant growth and development (*Dąbrowska, Prusińska & Goc, 2006*; *Boniecka et al., 2017*) The data about the expression of *RSH* genes during the early stages of seedling development are rather scarce. It was shown that during early seedling growth of *A. thaliana*, *RSH2* and *RSH3* are more strongly expressed than *RSH1* and *CRSH* (*Schmid et al., 2005*; *Mizusawa, Masuda & Ohta, 2008*; *Sugliani et al., 2016*). Interestingly, 8-day-old seedlings of *A. thaliana* *RSH3*-overexpression lines showed dwarf chloroplasts, metabolite reduction, and significantly inhibited plastid translation and transcription (*Maekawa et al., 2015*). Previously, we showed that, in the presence of plant growth-promoting rhizobacteria, the expression level of *BnRSH* genes in canola seedlings significantly raised (*Dąbrowska et al., 2021b*), so it was surprising that in this study *BnRSH1–3* and *BnCRSH* genes were unaffected or significantly downregulated in several experimental variants, including ones with *T. viride* (Fig. 3). Probably, the response to beneficial fungi is executed *via* different pathways or has different pacing than the response to beneficial bacteria. Importantly, we did not observe any abnormalities in the early stages of seedling development (File S2). In *I. nil* seedlings, *InRSH*s showed dynamic expression patterns during the early stages of seedling growth. During the 1-day sampling period, the expression level increase for *InRSH1*, decreased for *InRSH2*, and for *InCRSH* remained unchanged (*Prusińska et al., 2019*), showing that *RSH*s are differentially involved in seedling growth, not only plant-PGPR interaction.

Our previous *in silico* promoter analysis of the *BnRSH* genes revealed the presence of biotic stress response elements only within *BnRSH1* and *BnCRSH* promoters (*Dąbrowska et al., 2021b*). *BnCRSH* harbors AT-rich sequence elements for fungal elicitor-mediated activation, while *BnRSH1* contains both W-box (WRKY binding site) and TC-rich repeats elements involved in wounding and pathogens response (*Diaz-De-Leon & Lagrimini, 1993*; *Dąbrowska et al., 2021b*). WRKY, a plant-specific transcription factor family, plays vital roles in pathogen defense, abiotic stress, and phytohormone signaling (*Jones & Dangl, 2006*). It is also engaged in plant growth and development (*Chen & Yin, 2017*). Our results demonstrated that *T. viride* used in seed coating was not perceived by the plant as a pathogen and did not trigger an alarmones-dependent stress response pathway. Moreover, chitin and methylcellulose also did not induce the expression of *BnRSH* genes (Fig. 3). If plant immunity established *via Trichoderma* inoculation involves the alarmones-dependent pathway, we would expect that, in the presence of plant pathogens,

the expression profile of *BnRSHs* would change, probably with higher amplitude for *Trichoderma*-coated seeds. However, this hypothesis needs to be verified in the future.

Superoxide dismutase is one of the first-line antioxidant enzymes and is localized in various subcellular structures. Many stresses accelerate its activity (*e.g.*, salinity (*Houmani et al., 2016*), drought (*Saed-Moucheshi et al., 2021*), heat (*Ji et al., 2021*), and plant pathogens (*Gajera et al., 2016*; *Lightfoot, Mcgrann & Able, 2017*), and thus it is often chosen as an indicator of plant stress. Previous studies showed that fungi belonging to *Trichoderma* increased SOD activity in unstressed plants and plants under stress conditions. In tomato, SOD activity was elevated by the presence of *T. harizanum* in control conditions and the increase was even more profound upon osmotic stress treatment (*Mastouri, Björkman & Harman, 2012*). The presence of *Trichoderma longibrachiatum* Rifai induced SOD activity in wheat seedlings in the control conditions and higher SOD activity was observed in seedlings inoculated with fungi under saline treatment (*Zhang, Gan & Xu, 2016*). In groundnut, the activity of SOD was significantly induced by the presence of the pathogen *Aspergillus niger*, and the co-inoculation of plants with *A. niger* and *T. viride* led to a greater increase in SOD activity (*Gajera et al., 2016*). Interestingly, the seedlings displayed a slight but significant increase in SOD activity only when grown from seeds coated exclusively with methylcellulose and not in other variants—even those containing this substance (Fig. 4). Such a situation may be due to the effect of the crosstalk between methylcellulose, chitin, and *T. viride* within a complex regulatory network that leads to an unchanged SOD activity level. Although methylcellulose is an inhibitor of cellulase that prevents the cellulolytic activity of pathogenic fungi (*Cheng et al., 1991*), and it is a potential signal of pathogen invasion, the higher SOD activity is not necessarily a sign of oxidative stress defined as an imbalance between ROS production and scavenging. The increase in SOD activity and/or expression of genes encoding SOD was found at early seedling development stages in many plant species, including soybean (*Puntarulo et al., 1991*; *Gidrol et al., 1994*), goosefoot (*Bogdanović, Radotić & Mitrović, 2008*), and mung bean (*Singh, Chaudhuri & Kar, 2014*). ROS homeostasis does not mean that the level of ROS remains unchanged throughout the plant ontogenesis but, rather, that its level is adjusted to the current developmental and environmental context. For example, during the germination of tomato seeds, the level of superoxide anion dramatically rises, while the activity of SOD is maintained at a constant level (*Anand et al., 2019*). This event cannot be regarded as oxidative stress but rather as an oxidative burst because it is closely related to the plan of plant development. Moreover, promoters of genes encoding SOD enzymes harbor *cis*-elements involved in response to hormones and light (*Feng et al., 2016*; *Wang et al., 2017*; *Huo et al., 2022*). This implies that the expression of *SOD* genes might be regulated by developmental-related factors.

In *Arabidopsis*, loss-of-function mutation of chloroplast Cu/ZnSOD-encoding gene results in significant inhibition of plant growth and development and decreased chloroplast size, chlorophyll content, and photosynthetic activity compared with the wild-type plant (*Rizhsky, Liang & Mittler, 2003*). The *A. thaliana fsd1* knockout mutant, lacking functional FeSOD, extends fewer lateral roots than the WT strain (*Dvořák et al., 2021*).

As we already mentioned, *Trichoderma* was found to induce systemic resistance during biotic stress (*Kthiri et al., 2020*). Thus, we would expect significant differences in the activity of antioxidant enzymes between fully coated and uncoated seeds in the presence of a stressor rather than in plants grown in optimal conditions. This hypothesis is worth exploring, and our future experiments will focus on verifying it.

## CONCLUSIONS

We conclude that the complete products that contain methylcellulose, chitin, and spores of *T. viride* strain I and II outperformed other variants in different ways. *T. viride* strain I had greater potential to inhibit the growth of plant pathogens, *i.e.*, *B. cinerea*, *F. culmorum*, and *Colletotrichum* sp., whereas *T. viride* strain II was slightly more effective in the promotion of plant growth. The key features of the seed coating obtained in this study are: (1) the ability to promote seedling growth and limit the growth of plant pathogens contained in a single product, and (2) seed coating is safe for the environment because it is fully biodegradable. The multidimensional positive effects of our seed coating make it of great interest for sustainable agriculture. The developed seed coating is currently being tested in field conditions.

## ACKNOWLEDGEMENTS

The authors wish to thank prof. dr hab. Katarzyna Hrynkiewicz and dr hab. Maria Swiontek Brzezinska, prof. NCU for providing the fungus strains used in this study.

### Funding

This work was supported by Nicolaus Copernicus University under the Excellence Initiative-Research University programme (IDUB), competition "Grants4NCUStudents". The funders had no role in study design, data collection and analysis, decision to publish, or preparation of the manuscript.

### Grant Disclosures

The following grant information was disclosed by the authors:
Nicolaus Copernicus University under Excellence Initiative—Research University Programme (IDUB): "Grants4NCUStudents".

### Competing Interests

The authors declare that they have no competing interests.

### Author Contributions

- Sena Turkan performed the experiments, analyzed the data, prepared figures and/or tables, authored or reviewed drafts of the article, and approved the final draft.
- Agnieszka Mierek-Adamska analyzed the data, authored or reviewed drafts of the article, and approved the final draft.
- Milena Kulasek performed the experiments, analyzed the data, prepared figures and/or tables, authored or reviewed drafts of the article, and approved the final draft.
- Wiktoria B Konieczna performed the experiments, analyzed the data, authored or reviewed drafts of the article, and approved the final draft.
- Grażyna B Dąbrowska conceived and designed the experiments, analyzed the data, authored or reviewed drafts of the article, and approved the final draft.

## Data Availability

The data is available at Zenodo: Sena Turkan, Agnieszka Mierek-Adamska, Milena Kulasek, Wiktoria Beata Konieczna, & Grażyna Barbara Dąbrowska. (2022). Seed coating for *Brassica napus* L. [Data set]. Zenodo. https://doi.org/10.5281/zenodo.7628268.

## Supplemental Information

Supplemental information for this article can be found online at http://dx.doi.org/10.7717/peerj.15392#supplemental-information.

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
