# Peer review of "New seed coating containing Trichoderma viride with anti-pathogenic properties"

_PeerJ, doi:10.7717/peerj.15392_

## Round 0.1 · original submission · Major Revisions

Both reviewers and I found this paper to be scientifically significant as well as practical. However, there are still many issues that need to be revised, especially the detail description of the experimental method; the analysis, comparison, and discussion of the data. The reviewers also gave a lot of detailed revision comments.

Therefore, I suggest you make serious and comprehensive revisions to this paper.

Reviewer 1 ·

Basic reporting

Dear Authors,
The presented manuscript describes a study on coatings of canola seeds, their effects on plant growth and their safety for plants. Additionally, as seeds were coated with living microorganisms, their inhibitory effect on the three pathogens was evaluated.
I believe that this study is interesting and there is some cool biology within it. It can be extended in many ways, but I decided to focus on parameters that were already measured. I believe that the selection of tested parameters is adequate. However, there are some major issues with the measurements (inhibitory effect and ability of seeds to complete germination) – fortunately it can be probably recalculated from the existing data. Additionally, statistical analyses were not described. I also made some detailed comments on data presentation (although I think that Figure 3 and 4 are of very good quality).
Summarizing, I believe that this manuscript is worth further revision but it needs a lot of attention and some weighty changes. Thus, I recommend to subject it for major revision.
List of all major issues and detailed line-by-line comments are appended in 'Additional comments' section.

Experimental design

Overall, experimental design meets standards of PeerJ. List of requested improvements is appended in section 'Additional Comments'.

Validity of the findings

Overall, the findings are validate and meets standards of PeerJ. List of requested improvements is appended in section 'Additional Comments'. In general, some measurements should be improved (request for recalculation of data) and some fragments in 'Discussion' should be reworked.

Additional comments

MAJOR ISSUES

Statistical analysis is not described in the ‘Material and Methods” section. Please provide information about test of normality, variances, tests used for comparisons and used post hoc test. Figures 1 and 2 as well as Table 2 are missing statistical indicators and proper descriptions. Please provide which differences were significant in ‘Result’ section.

Please provide a table summarizing all the studied variants. You can see an example in Skwarek et al. (2021). It is hard to see any details on each variant in the current version of the manuscript. Please provide explanation of all abbreviations. You can also provide weights of coated and uncoated seeds.

Request for global change: correct use of germination-related terms. Please replace all terms “seeds germinated” for “seeds completed germination” as we do not know if seeds that did not completed germination germinated or not, until, e.g., embryo measurements are conducted. Please use “germinated” only in context of process before protrusion of testa. Check where applicable (see e.g., L357).

Measurements of pathogen inhibition: for more details, see comment on the L

Please provide list of 3-4 detailed hypotheses that you tested in this study and list of associated questions at the end of ‘Introduction’.

I truly miss take-home message at the end of ‘Discussion’. Please state which coating is the best, which one is the worst, implications for usage of such coatings as well as some further scientific directions in this topic.

Raw data: probably it is not mandatory, but I believe that.xls file is easier for quick calculations or data inspection than.csv. This is for decision of the Authors and the Editor.

DETAILED LIST OF COMMENTS (LINE-BY-LINE)

L76: Replace “confirms” with “strongly suggest”
L91: Double parentheses; please remove it
L94: Symbol for superoxide radical is not correct
L110: Please gove full name of species, namely, Pharbitis nil (L.) Choisy, or (better) “Ipomoea nil (L.) Roth (syn. Pharbitis nil (L.) Choisy)”
L117: In which one Arabidopsis?
L135-136: You can apply full names for names of fungi as well, e.g., Botrytis cinerea Pers. (please use Index Fungorum for current nomenclature). Additionally, check Trichoderma viride in the same manner.
L161: I doubt about method you used for quantification of inhibition. Is determination of radius a good estimator in the case when the colonies are not circular? I see that is a little problematic to determine sharp boundary of each colony. However, maybe it could be better to compare areas (ratio between control and inhibition assay)? Anyway, please provide a strong reference for this issue.
L165: Please use correct nomenclature for naming cultivars, i.e., ‘Karo’ but not cv. Karo. Additionally, please add some more information about this cultivar. What was the reason to use this cultivar for experimentations? Is this a cultivar registered in Poland? Is it hybrid, or nonhybrid?
L166: Double parentheses; please remove it
L168: It could be better to use SI unit, thus 0.4 cm-3 is better choice. Check all similar cases throughout the manuscript and make corrections where needed (i.e., milliliters/microliters/liters to dm3/cm3)
L171-L172: Please provide more details about procedure of coating. It was manual coating or any device was used?
L178: Please provide water quality grade or its conductivity
L178: Were the dishes sealed, e.g., with parafilm?
L178: Please provide PAR value and light source in germination cabinet
L178: How many dishes containing 25 seeds were used per single variant? This question is also associated with missing standard deviation bars in Figure 2.
L178: This comment is associated with data you presented in Figure 2. I suggest to calculate both final germination percentage (FGP) and index of germination velocity (IGV; known also as Timson’s index) as you counted germination indices daily. For more details on the methodology of calculations, please see Khan and Ungar (1997) and Ranal and Santana (2006).
L179-180: Please provide more details on measurements of length as they are not easy considering morphology of seedlings from Petri dishes.
L180: How many seedlings from single dish was probed? Were they similar in their size?
L187-188: Please provide plant material to extraction buffer ratio
L193: I believe that the original method for the determination of SOD activity is from Beauchamp and Fridovich (1971) or a similar source, with modifications of Rusaczonek et al. (2015), as all the main ingredients for determination are the same.
L282: Double parentheses; please remove it
L301: I believe that you mean seed coating but not seed coat. Please correct where possible.
L301-306: It better fits to the ‘Material and Methods’ section. Please rework this.
L324: Please see comments for Figure 1; in this context, please add some information about differences between strains and pathogens. Right now, L324-355 is too descriptive.
L357: This is a good example: “tested strains of T. viride stimulated ability of seed to complete germination” but not “tested strains of T. viride stimulated seed germination”. Please see my third major comment. Check and correct where applicable.

Table 2: In my opinion, standard deviation, but not standard error should be presented. I also suggest conversion of this table into three (or six) separate subfigures. Statistical indicators are missing. In this setup, it is very reasonable (and mandatory) to compare all the studied variants with one-way ANOVA. Mandatory comment: Please calculate Hypocotyl/Root Dry Matter Content (mg DW x mg-1 FW) – I believe that there can be some weighty differences between variants, e.g., M vs M+C+T. Additionally, please calculate fresh biomass and dry biomass hypocotyl to root ratio.

Figure 1: I feel that this figure could be slightly simplified but it is not mandatory. Please be aware of recommendation of figure quality for production purposes. However, more data can be extracted from panel A of this figure. Please provide mean and standard deviation for inhibition of the studied pathogens by the studied Trichoderma. Additionally, please analyze such data with two-way ANOVA as it is very valuable, totally possible and very informative (factor 1: Trichoderma I/II and factor 2: pathogen species) – provide F values, significance and differences.

Figure 2: It looks very nice but it is hard to follow. More suggestion on this issue is provided in detailed comment for line 178.

Figure 3 and 4: Very good quality and description. Please use similar manner (e.g., letter-based statistical indicators) in all the other figures and tables.

References:

• Ranal, M.A., Santana, D.G.D.E., 2006. How and why to measure the germination process? Revista Brasil. Bot. 29, 1–11.
• Khan, M.A., Ungar, I.A., 1997. Effects of thermoperiod on recovery of seed germination of halophytes from saline conditions. Am. J. Bot. 84, 279–283.
• Skwarek, M., Wala, M., Kołodziejek, J., Sieczyńska, K., Lasoń-Rydel, M., Ławińska, K., Obraniak, A., 2021. Seed Coating with Biowaste Materials and Biocides—Environment-Friendly Biostimulation or Threat? Agronomy (Basel) 11, 1034.

Reviewer 2 ·

Basic reporting

The manuscript describes a study on the preparation of a seed coating containing the spores of Trichoderma strains and assessment of its effect on the seed germination, seedling growth and stress response in Brassica napus. Improvements need to be made in many sections:

1. The word "novel" in the title is not suitable as the production of seed coats with fungal spores have been extensively reported previously.

2. The introduction section is too long and does not provide a clear rationale of the study. Suggest to remove unnecessary details, instead, focus on the research gap.

3. Similarly, the discussion is too long and written in a rather general manner. There is a lot of factual information on, for instance, mechanisms of action of Trichoderma (lines 325-355) that was not investigated in this study.

4. Please ensure information of the replicates (technical or biological) is clearly presented for all tables and figures.

5. Tables and figures should be able to stand alone, hence, please explain all abbreviations used in all tables and figures.

6. The conclusion does not answer the objectives of the study. Are all objectives met?

7. Please check the language used throughout the manuscript, for example, please refer to line 233.

8. There are other mistakes in the manuscript, for example, canola (Brassica...) in line 14. Please check the manuscript thoroughly.

This part fails to meet the standard of PeerJ.

Experimental design

Standard methodologies were used. Sufficient details were provided for most parts. Nonetheless, improvements are needed:

1. Section 2.2: Is the 6-days period suitable considering all fungi have different growth rate? Would the cultures be overgrown after 6 days?

2. Please state the source of the Brassica napus seeds, and cultures of the plant pathogenic fungi.

3. Section 2.5: Why did the authors choose the determine the level of SOD and not other antioxidant enzymes?

Validity of the findings

Trichoderma strains used in this study was shown to inhibit the growth of selected pathogens and hence was chosen to be included in the coat. The seed coats were shown to have beneficial effects on germination and growth. The main issue is the lack of analysis and discussion on the data of this study, and if the findings are comparable to those in the literature. There is simply insufficient comparison to previous work.

Some suggestions for improvement:

1. The inhibitory effect of the Trichoderma strains (quantitative data) should be presented in the form of a bar chart. What is the justification for Fig 1B?

2. Suggest to discuss the data obtained in this study with those from related studies to make better conclusions regarding the findings from this study. Many aspects have not been discussed such how the composition of the coat may affect its properties as observed in the germination and growth tests.

Additional comments

The subject matter of this study seems interesting and might have potential application in the agriculture sector but findings are still preliminary in nature. More characterisation is necessary. Data presentation, analysis and interpretation have to be improved based on the above comments. The focus of the manuscript is not clear. The authors are suggested to remove the unnecessary information.

---

## Round 0.2 · Minor Revisions

The reviewers and I agreed that the paper had been well revised and basically met the requirements for publication. However, there are still some minor issues that need to be improved, so please follow the reviewers' comments and revise carefully.

Reviewer 1 ·

Basic reporting

Dear Authors,
Thank you for your response. I rate your improvements as adequate with some expections. In my opinion, the manuscript should be subjected to minor review.

Experimental design

Experimental design meets standards of PeerJ.

Validity of the findings

Findings are validate with slight expections - some data from seedlings is discussed in the context of germination. Detailed informations are provided in 'Additional comments'.

Additional comments

General comment: Believe that using of cm^-3 is not wrong, it is very desirable. I understand that for common practice mL or µL are used. However, when you select one convention it should be followed thought the for 20 µL you can use 0.02 cm^-3 or 0.002 cm^3 for 2 µL (if you wish not to use mm^-3). I know that the common practice to write all volumes below 1 cm3 as µL but I suggest to do this better than mediocre work – it is correct, not weird.

L33: How it enhanced germination ratio? In my opinion, the main finding is that the tested coatings did not negatively affected ability of the seeds to complete germination. Please compare with Table 3. It is well described in ‘Results’. Please note that the high SD value for FGP under the control treatment probably caused this result.
L162: H2O but not H20, please check where possible
L388: Please add source for this: “Nowadays, they are made mostly from synthetic
389 materials that are expensive and non-degradable.”
L389-391: As seed complete germination but not germinate (in this case).
L392-397: This fragment is very general and does not fit to ‘Discussion’. Please relocate it into ‘Introduction’ or delete it.
L413: ‘Morden’ or Morden?
L435-436: I do not agree, as the studied treatments did not affected germination. On the other hand, you can state that it does not affect germination in a negative way.
L443: Improved ability of seeds to complete germination
L449: higher germination: if you mean higher FGP values, it is better to state: “Higher ability of seeds to complete germination…” or “Higer FGP values…”.
L469-474: Most likely, this work can be useful for you (review on Trichoderma; Nawrocka and Małolepsza, 2013; doi: 10.1016/j.biocontrol.2013.07.005).
L478-480: I believe that you mean early growth of seedling as completion of germination is equal to protrusion of testa by embryo. Thus, the process you describe is not ‘early germination’ but early growth of seedling. Early germination is associated with imbibition. I do not state that elements, compounds and organism do not affect imbibition, but you clearly referred to effects of microorganism on colonization of roots.
L484-492: It is a mix of conventions. Your data pertains to expression of genes in seedlings and you discuss it in the context of germination. I believe that it should be discussed in the context of early growth of seedlings. Please compare with 1-2 other works on expression of these genes at early growth of seedlings.
L508-509: The audience need to know how SOD acts upon colonization by Trichoderma and by pathogens, but not during drought or under saline stress. Please find and cite such works.
L517-518: Please provide evidence that higher activity of SOD can be effect of other stimuli but not developing stress.
L518-526: You measured activity of SOD in seedlings and discuss it in the context of germination, seed longevity and vitality. Please change it; you can use the storyline that the tested treatments were not stressful from germination to growth of seedlings which was reflected in the activity of SOD being marker of early stress response. Please note that you did not measured seed vitality!
L538-539: Please remove “In the future, it would be possible to use the seed coating containing T. viride strains. The new seed coatings are currently tested in the field conditions.” On the other hand, it would be of interest to answer the following questions: Is the effects of coating multilayer, what is the key aspect of Trichoderma coatings: alternation of growth of competition with pathogens, is the effect stronger for germination or early growth of seedlings, is it safe?

Reviewer 2 ·

Basic reporting

The authors have provided satisfactory response to my queries. I am in agreement that "new" is a more suitable word to replace "novel" in the title.

Experimental design

No comment

Validity of the findings

No comment

---

## Round 0.3 · Minor Revisions

I have read your response and manuscript carefully and feel that there are proper explanations for the reviewer's questions, so I am almost ready to accept this article.

But I strongly suggest that next time when you answer the comments of reviewers, don't just answer "was edited" and "were rewritten". It's better to answer how it was revised and explained, which is more convenient for reviewers and editors to evaluate.

Before I can accept the manuscript, please address the following comments from the Section Editor:

> Proofreading is necessary before acceptance since the grammar is often wrong and some word choices are odd. As well, the figure colors should be optimized for color-blind readers.

---

## Round 0.4 · accepted · Accept

I believe you have tried your best to improve the language and the quality of the pictures. The revision is effective. I think this article is acceptable to be published. Congratulations!